# Ubiquitous strong electron–phonon coupling at the interface of FeSe/SrTiO₃

Chaofan Zhang[1,2], Zhongkai Liu[3], Zhuoyu Chen[1,2], Yanwu Xie[1,2], Ruihua He[4], Shujie Tang[1,2], Junfeng He[1,2], Wei Li[1,2], Tao Jia[1,2], Slavko N. Rebec[1,2], Eric Yue Ma[1,2], Hao Yan[1,2], Makoto Hashimoto[5], Donghui Lu[5], Sung-Kwan Mo[6], Yasuyuki Hikita[1], Robert G. Moore[1,2], Harold Y. Hwang[1,2], Dunghai Lee[7] & Zhixun Shen[1,2]

The observation of replica bands in single-unit-cell FeSe on SrTiO₃ (STO)(001) by angle-resolved photoemission spectroscopy (ARPES) has led to the conjecture that the coupling between FeSe electrons and the STO phonons are responsible for the enhancement of $T_c$ over other FeSe-based superconductors. However the recent observation of a similar superconducting gap in single-unit-cell FeSe/STO(110) raised the question of whether a similar mechanism applies. Here we report the ARPES study of the electronic structure of FeSe/STO(110). Similar to the results in FeSe/STO(001), clear replica bands are observed. We also present a comparative study of STO(001) and STO(110) bare surfaces, and observe similar replica bands separated by approximately the same energy, indicating this coupling is a generic feature of the STO surfaces and interfaces. Our findings suggest that the large superconducting gaps observed in FeSe films grown on different STO surface terminations are likely enhanced by a common mechanism.

[1] Stanford Institute for Materials and Energy Sciences, SLAC National Accelerator Laboratory, 2575 Sand Hill Road, Menlo Park, California 94025, USA. [2] Geballe Laboratory for Advanced Materials, Departments of Physics and Applied Physics, Stanford University, Stanford, California 94305, USA. [3] School of Physical Science and Technology, ShanghaiTech University, Shanghai 200031, China. [4] Department of Physics, Boston College, Chestnut Hill, Massachusetts 02467, USA. [5] Stanford Synchrotron Radiation Lightsource, SLAC National Accelerator Laboratory, 2575 Sand Hill Road, Menlo Park, California 94025, USA. [6] Advanced Light Source, Lawrence Berkeley National Laboratory, Berkeley, California 94720, USA. [7] Department of Physics, University of California at Berkeley, Berkeley, California 94720, USA. Correspondence and requests for materials should be addressed to Z.S. (email: zxshen@stanford.edu).

The discovery of high-temperature superconductivity in single-unit-cell (1 UC) FeSe/STO(001) continues to attract a great deal of interest[1]. In particular, a recent mutual inductance measurement has shown an onset of diamagnetism at the same temperature (65 K) as when a gap emerges in angle-resolved photoemission spectroscopy (ARPES), thus supporting the superconducting origin of the observed single-particle energy gap[2]. Compared with the 8 K superconducting transition temperature of bulk FeSe the $T_c$ of the 1 UC FeSe/STO(001) is almost an order of magnitude higher. This leads to a natural question regarding the cause of the $T_c$ enhancement.

Apparently, one contributing factor is electronic doping. Earlier works[3–5] on intercalated $A_x Fe_{2-y} Se_2$ (A = K, Tl, Cs, Rb and so on) compounds have shown superconductivity with $T_c$ above 30 K. Recent work on a bulk crystal of $Li_{1-x} Fe_x OHFeSe$, consisting of FeSe layers intercalated with $Li_{1-x} Fe_x OH$ shows $T_c \sim 41$ K (refs 6,7). On a different front, when the top surface of a non-superconducting 3 UC FeSe/STO(001) is coated with potassium (K), a superconducting gap is observed at 48 K (ref. 8). However, despite the similarity in the fermiology of these systems with that of the 1UC FeSe/STO(001), the $T_c$ of the later is still significantly higher. This suggests that electron doping alone is insufficient to account for the full enhancement of $T_c$ in 1 UC FeSe/STO(001).

The foremost telling clue concerning the origin of the extra $T_c$ enhancement in 1 UC FeSe/STO(001) comes from the ARPES observation of 'replica bands' at $\sim 100$ meV below the main band[9]. Such replica bands are explained in terms of a 'shake off' phenomenon—in the photoemission process of ejecting an electron, part of the incoming photon energy can be used to excite a vibration quantum. Thus, the replica bands signify a strong coupling between the FeSe electrons and STO phonons, which in turn is conjectured to be the $T_c$ enhancement mechanism[9].

In two very recent publications scanning tunnelling microscopy and ARPES reported similar superconducting gap and gap closing temperature of 1 UC FeSe grown on the (110) surface of STO[10,11]. Since the (110) surface is geometrically and chemically different from that of (001) surface, it raises doubts[11,12] whether the same electron–phonon-enhancement mechanism also applies.

We conducted ARPES studies of FeSe films on STO(110) to investigate this issue. In addition, we have carried out a comparative study of the electron–phonon coupling on bare STO(001) and STO(110) surfaces. Similar electron–phonon replica bands have been found for the two-dimensional electron gas formed on the (001) surface[13,14]. Our results show that the electron–phonon replicas at the FeSe/STO(110) interface and bare STO(110) surface are similar to its counterparts at the (001) surface, suggesting such electron–phonon coupling is a generic feature of STO surfaces and interfaces prepared under certain conditions. As such, the recent observation in the FeSe/STO(110) system is consistent with the conjecture of $T_c$ enhancement by interface electron–phonon coupling.

## Results

**Replica bands of 1UC FeSe grown on STO(110) and STO(001).** One important difference between the interfaces of FeSe/STO(110) and FeSe/STO(001) is the distortion of the in-plane unit cell from tetragonal to orthorhombic symmetry. A comparative ARPES study of FeSe/STO(110) and FeSe/STO(001) has been recently reported by Zhang et al.[11] However, the replica bands were not resolved in their experiment. By utilizing the capability of in situ MBE growth and high-resolution ARPES beamline, we have carefully studied the band strutures of the 1UC FeSe/STO(110) system. Our results show the hole band at Gamma point shifts towards lower binding energy (with the band top located at $\sim 20$ meV below Fermi level) compared with FeSe/STO(001) (see Supplementary Fig. 1). It has been reported that the presence of incipient bands below $E_F$ may have an important role in the electron pairing[15]. Figure 1a shows a Fermi surface at the M point of the Brillouin zone measured at $\sim 25$ K. Figure 1b shows the energy–momentum intensity map along a high-symmetry cut through M. It clearly shows an electron band crossing the Fermi level. The superconducting gap opening at the Fermi momentum ($k_F$) is measured to be

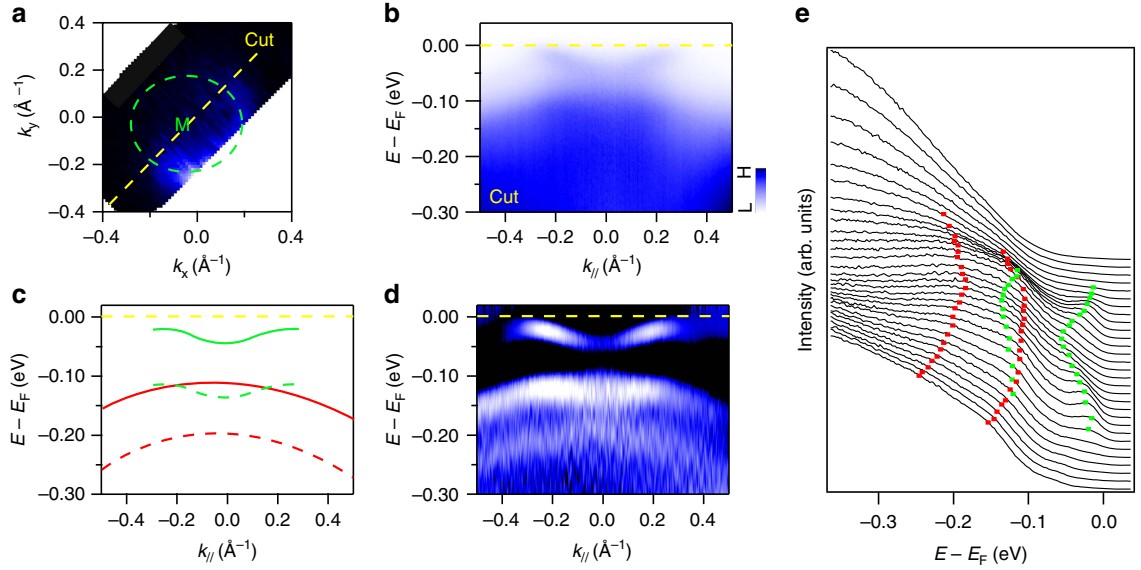

**Figure 1 | The ARPES spectra of 1UC FeSe grown on STO(110).** (**a**) The M point Fermi surface map of FeSe/STO(110). The yellow dashed line shows the high-symmetry cut along Γ-M direction. (**b**) Energy–momentum intensity map of FeSe/STO(110) around the cut marked at **a**. (**c**) Schematic representation of the electron band (green) and hole band (red) of FeSe/STO(110). The replica bands are shown as the dashed lines. (**d**) Second derivative image of (**b**). Comparison with **c** identifies the features associated with the main bands and the replica bands. (**e**) EDCs near M shown as a waterfall plot with main and replica bands marked by corresponding colour squares.

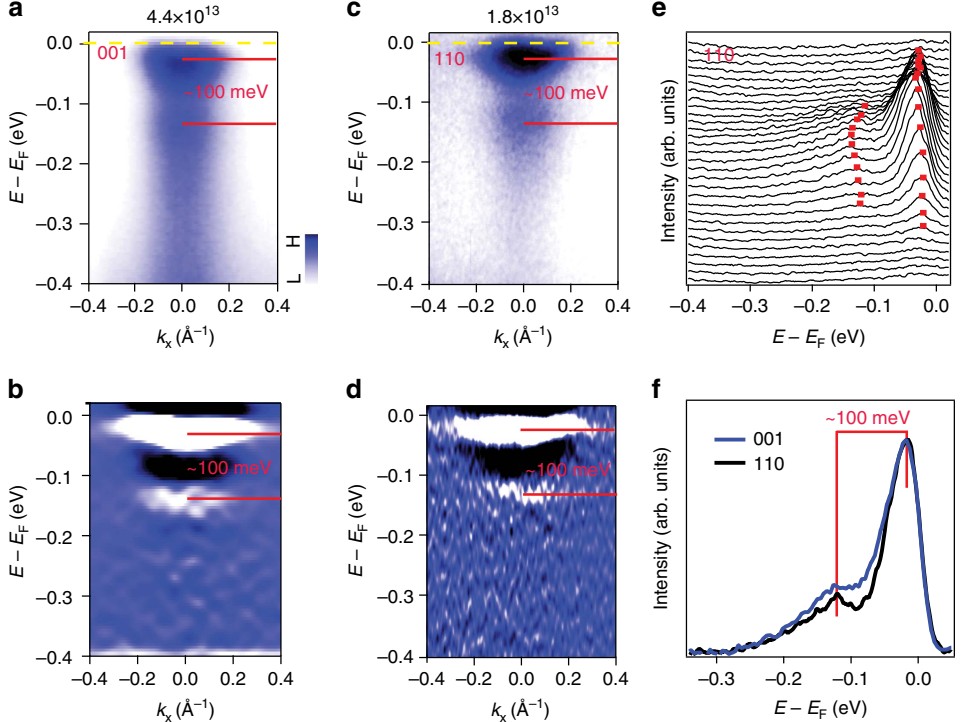

**Figure 2 | Comparison of the ARPES spectra for the surface bands at STO(001) and STO(110).** (**a**,**c**) Energy–momentum intensity map of STO(001) and STO(110) with similar main bands and replica bands. The separation of two bands is estimated to be ∼100 meV in both surfaces (**b**,**d**) Second energy derivative of **a**,**c**. (**e**) EDCs of STO(110) shown as a waterfall plot with main and replica bands marked by red squares. (**f**) EDCs of STO(001) and STO(110) measured at $k_F$, showing the similar ∼100 meV energy separation between the main and replica bands.

∼14 meV, similar to that observed in FeSe/STO(001) (ref. 9). To better resolve the band dispersion, we show the second energy derivative of the ARPES intensity in Fig. 1d, and mark the peaks of the energy distribution curves (EDCs) with green and red squares in Fig. 1e. In Fig. 1d,e we see both an electron band crossing the Fermi level and a hole band below it. Most interestingly, at ∼100 meV below the main bands, we again see the features that can be identified as the replica bands. The replica of the hole band (red dash line) is clearly presented at Fig. 1d, the totality of the data in Fig. 1d,e also makes the replica of the electron band (green dashed line) discernable. Sketches of the main bands and replica bands are shown by the solid and dashed curves, respectively in Fig. 1c. Thus replica bands with similar energy separation as in 1UC FeSe/STO(001) have been clearly identified in FeSe/STO(110).

**Surface electronic structure of STO(110) and STO(001).** To lend further support for the similarity of electron–phonon coupling on these surfaces and interfaces, we proceed with a comparative study the surface electronic structure of pure STO(110), in reference to the (001) surface where the replica band has been recently reported[13,14]. The two-dimensional electron gas (2DEG) created by exposing the STO(001) surface to synchrotron radiation in ultrahigh vacuum, that is, photodoping, has been extensively studied[16–18]. By tuning the dosage of ultra violet exposure, which increases the electron density, and by adding atomic oxygen, which decreases the electron density, very precise control of the 2DEG density can be achieved[19,20]. Similar to STO(001), a 2DEG has also been observed on the (110) (ref. 21) and (111) (ref. 20) surfaces of STO. In particular, ARPES has observed the photo-doped surface conduction bands as well as their quantum well subbands due to the spatial confinement.

The STO samples used in this study are from the same source as the STO(001) substrates upon which we deposited the 1UC FeSe films in ref. 9. Our results on the surface electronic structure are consistent with those reported by Wang *et al.*[21], as shown in the Supplementary Fig. 2. We focus our discussion on lower photodoping density regime, where the replica is most visible. In Fig. 2a,c we compare the APRES data of STO(001) and STO(110) at electron density of $n_{2D} \approx 4.4 \times 10^{13}$ and $1.8 \times 10^{13}$ cm$^{-2}$, respectively. These electron densities are chosen so that the $k_F$ of the main bands are similar for the two surface terminations (a schematic of the two Fermi surface is presented in supplementary Fig. 3). In this doping regime we can only resolve the parabolic $d_{xy}$ main bands as the quantum well subbands cannot be resolved at this doping level. Importantly, a band with similar effective mass at higher binding energy (∼100 meV) can also be resolved for both surface terminations. This is the feature we identify as the replica band. To enhance the contrast we plot the energy second derivative of the ARPES intensity in (b and d). The EDCs of STO(110) have also been shown as a waterfall plot in Fig. 2e to highlight the main band and the replica band. For both cases we determine the energy separation between the main band and the replica band to be ∼100 meV. To more accurately estimate the energy separation and compare the relative intensity between the replica and main bands we show the EDCs at $k_F$ in Fig. 2f. We have normalized the intensities, so that the peaks associated with the main bands coincide for the two surface terminations. The deduced energy separation between the main band and the replica band are the same, within the experimental error.

**ARPES study of STO(110) as a function of photodoping.** To further understand the electron–phonon coupling on STO surfaces, we explore its doping dependence. Figure 3 shows the

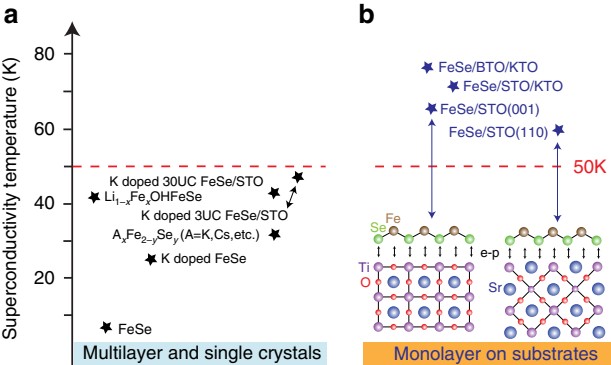

**Figure 3 | The photodoping evolution of the ARPES spectra for the surface bands of STO(110).** (**a**–**h**) The evolution of the ARPES dispersion as the surface carrier density increases from $n_{2D} \approx 1.8 \times 10^{13}$–$6.0 \times 10^{13} \, cm^{-2}$. The red curve marks the replica band in **a**, while the yellow curve marks quantum well subband in **h**. The upper part of each panel shows the second energy derivative of the lower part in the $-100 \, meV \leqslant E\text{-}E_F \leqslant 0$ energy window. (**i**) EDCs of the main band and replica band as the carrier increases from $n_{2D} \approx 1.8 \times 10^{13}$–$6.0 \times 10^{13} \, cm^{-2}$. The dashed red line marked the positon of replica band.

**Figure 4 | Superconductivity temperature of FeSe-related superconductors.** (**a**) All the multilayer and single crystals of iron-based superconductors show $T_c$ lower than 50 K. (**b**) In contrast, monolayer FeSe on various TiO$_2$ terminated substrates always show $T_c$ higher than 50 K. We have observed electron–phonon coupling both at FeSe/STO(001) and FeSe/STO(110). The $T_c$ of FeSe/STO/KTO is from ref. 32.

band structure evolution of STO(110) with increasing photodoping. The carrier density increases from $n_{2D} \approx 1.8 \times 10^{13}$ to $6.0 \times 10^{13} \, cm^{-2}$. The upper part of each panel shows the energy second derivative of lower part in the energy range of $-100 \, meV \leqslant E - E_F \leqslant 0$. As the carrier density increases, the $d_{xy}$

band shifts towards higher binding energy, and its quantum well subband gradually appears, again resembling that of the STO(001) surface[14]. The subband is superimposed on the main band at first, then moves to lower binding energy and becomes clearly separated from the main band at higher carrier densities (the dashed yellow curve in Fig. 3h). Increased separation between the main and the subband with increased doping is consistent with the quantum well origin of the subbands. In contrast the replica band is better resolved at low carrier densities (see, for example, the red curve in Fig. 3a).

In Fig. 3i we show the doping dependence of the replica band intensity. Here the dashed red line marks the energy position of the replica band. We have estimated the intensity ratio between the replica and the main band as a function of the electron density as shown in Supplementary Fig. 4a. The carrier density dependence described above are similar to those observed by Wang *et al.*[14] on STO(001), consistent with a similar picture for electron–phonon coupling on STO surfaces. We also estimate the effective mass associated with the parabolic $d_{xy}$ bands at different carrier densities (Fig. 4b). If one attributes the change in the effective mass to the change in the electron–phonon coupling strength the result is consistent with the trend deduced from Supplementary Fig. 4a. However, it should be noted the change of effective mass has been used as a direct tool to reflect correlation effects[22]. The gradual doping of K on the top of FeSe single crystal shows a heavier effective mass with a higher $T_c$, which has been

interpreted as the enhanced correlations due to the development of a two-dimensional system on top of the single crystal.

We also observed a similar replica band with the same phonon mode on STO(111) surface at lower doping level (see Supplementary Fig. 5), again supporting the picture of universal behaviour of electron–phonon coupling in various STO surfaces and interfaces. This suggests the possibilities of high $T_c$ superconductivity in FeSe/STO(111) if good quality interfaces can be achieved.

## Discussion

The origin of high $T_c$ in 1UC FeSe/STO is still a question under investigation. From the experimental facts at least two contributing factors can be identified[23]: the electron doping and the substrate effect. The mechanism of electron doping at the interface of FeSe/STO(001) is still actively debated[9,24]. However the formation of oxygen vacancies may play an important role and have also been observed in FeSe/STO(110) (see Supplementary Fig. 7). The most direct evidence that electron doping raises $T_c$ is from the studies of potassium doping on otherwise non-superconducting multilayer FeSe films (or low $T_c$ bulk FeSe crystals). By coating the surface of bulk FeSe[22,25] and multilayer FeSe films[8,26–29] with potassium, $T_c$ can be increased to as high as 48 K, at which point the Fermi surfaces have a similar area as that of 1UC FeSe/STO, indicating approximately the same electron doping level. In addition, bulk materials where various donor layers are intercalated between the FeSe layers, for example, $A_xFe_{2-y}Se_2$ (ref. 30) and $Li_{1-x}Fe_xOHFeSe$ (ref. 6) also have similar Fermi surface volume and similar range of $T_c$. Nonetheless, the $T_c$ of these systems are still appreciably lower than 1UC films on STO or other titanate substrates[31] (see Fig. 4).

Thus it is natural to associate the extra enhancement of $T_c$ in the monolayer films to a substrate effect. In particular, the cross-interface coupling of the FeSe electrons and STO phonons has been proposed to enhance the $T_c$ of heavily electron-doped FeSe systems[9,22]. On the surface it is unusual that electrons in the top layer of K-doped 3UC FeSe/STO cannot couple to the STO phonons, that is, the electron–phonon coupling being very local.

To answer that question we recall that the doping in 1UC FeSe/STO is due to charge transfer between STO and FeSe, so that near the FeSe-STO interface there is an electric field. Such electric field will induce a layer of dipoles, so that near the interface with FeSe, STO is strongly polarized. In ref. 9,23 it has been argued that the phonons causing the replica are associated with the vibration of these dipoles with the displacement perpendicular to the interface, and the modulating wave vector parallel to interface. Such in-plane modulating dipole potential does not affect layers further away from the interface due to the screening by the electrons in the bottom FeSe layer. The fact that the bottom layer alone can screen the interface suggests that Coulomb attraction localizes nearly all the doped charges in the bottom FeSe layer. Thus while the top layer in K-doped 3UC FeSe/STO is equally electron-doped (by potassium rather than STO) it does not experience the modulating dipole potential due to the screening of other FeSe layers beneath it.

The above situation has a close analogy with what we observed for the doping dependence of the replica band on the surface of STO. As the photo-doped carrier density increases, the modulating dipole potential caused by the polar phonon is screened. As a result the replica band intensity vanishes.

In conclusion, we have observed the replica band for FeSe/STO(110) as well as lightly doped STO(110) surface. The similar energy separation between the main bands and replica bands for these interfaces/surfaces as those in FeSe/STO(001) and STO(001) suggest that similar electron–phonon coupling is involved in both. Our results suggest that a similar phonon-enhancement mechanism might also be at work for the superconducting FeSe/STO(110).

## Methods

**Samples.** The pure, undoped STO substrates were degassed at 450 °C for 1 h before *in situ* transfer to the ARPES measurement chamber. Single unit cell FeSe films were grown on 0.05 wt% Nb-doped STO(110) substrates. Before growth, the substrates were degassed at 450 °C for 45 min, and heated to 830 °C for 30 min in the ultra high vacuum (UHV) chamber and then held at $T_{sub} = 390$ °C for growth. FeSe was obtained by co-evaporating Fe and Se with a flux ratio of 1:10 and a growth rate of $\sim 2 UC \min^{-1}$. Post-annealing at 420 °C for 3 h was performed after growth.

**ARPES measurements.** All the measurements were performed *in situ* utilizing MBE + beamline ARPES systems in this work. The ARPES measurements of pure substrates were carried out at Beamline 10.0.1 of the Advanced Light Source of Lawrence Berkeley National Laboratory. The total energy resolution was $\sim 25$ meV with photon energy of 48 eV and a base pressure better than $5 \times 10^{-11}$ Torr. The measurements on 1UC FeSe/STO(110) films were performed at Beamline 5-4 of the Stanford Synchrotron Radiation Lightsource of SLAC National Accelerator Laboratory. The photon energy used was 24 eV with the total energy resolution $\sim 8$ meV and a base pressure also better than $5 \times 10^{-11}$ Torr.

**Data availability.** The data relevant to the findings of this study are available from the corresponding authors on reasonable request.

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

## Acknowledgements

The work at Stanford is supported by the US DOE, Office of Basic Energy Science, Division of Materials Science and Engineering, under award number DE-AC02-76SF00515. ALS and SSRL are supported by the Office of Basic Energy Sciences, US DOE under contract No. DE-AC02-05CH11231 and DE-AC02- 76SF00515, respectively. CFZ's postdoctoral fellowship is supported by Knut and Alice Wallenberg Foundation in Sweden. D.H.L. is supported by the US Department of Energy, Office of Science, Basic Energy Sciences, Materials Sciences and Engineering Division, grant DE-AC02-05CH11231.

## Author contributions

C.Z., Y.H., R.G.M., H.Y.H., D.L. and Z.S. proposed and designed the research. C.Z. and Z.L. carried out the ARPES measurements with help from Z.C., Y.X., J.H., S.T., T.J., W.L., H.Y. and S.N.R. S.-K.M., D.L., M.H. maintained the ARPES beamlines. C.Z. analysed the data with help from Z.L. and E.Y.M. C.Z. wrote the manuscript with D.L., R.G.M., R.H., H.Y.H. and Z.S. R.G.M., H.H., D.L., Z.S. are responsible for project direction and infrastructure. All authors discussed the results and commented on the manuscript.

## Additional information

**Competing financial interests:** The authors declare no competing financial interests.

**Publisher's note**: 

