## [Peer Review File · Nature Communications]

Reviewers' comments:

Reviewer #1 (Remarks to the Author):

The manuscript is indeed very interesting. The authors convincingly demonstrated the existence of the replica bands in FeSe/STO(001) and FeSe/STO(110), which play an important role in our understanding of the mechanism of T_c enhancement in these systems as compared to simple electron doping of the single layer.

1) One important point I would like the authors to address is the formation of the oxygen vacancies at the interface of STO and their relevance on the superconducting properties of the FeSe/STO interface. Formation of the vacancies is unavoidable in this system and the authors should comment on this.

2) Another important question concerns the position of the hole bands near the Gamma point. I think it is important to know their positions below E_F in ARPES in both FeSe/STO(001) and FeSe/STO(110) structures. If the strong coupling mechanism of superconductivity is involved the insipient bands below E_f may play also an important role.

Once these questions are addressed the manuscript could be recommended for publication

Reviewer #2 (Remarks to the Author):

Zhang et al. performed ARPES experiments on FeSe/STO(110), STO(110) and STO(001). They devised a careful experiment for a comparative study of FeSe thin film on STO(001) and STO(110). Citing the omnipresence of an energy loss mode whose energy is similar to phonon energy of ~ 100 mV, the authors argue that the phonons should play a critical role in enhancing the T_c . They provide a possible story for the strong coupling between the electron in FeSe and phonon in STO via formation of surface dipole. While observation of the mode which is likely to be from phonon does not necessarily mean the e-ph coupling is the culprit, the data and arguments are reasonable as a possible route to the enhanced superconductivity. The work further solidifies their original work that was published in Nature and clarifies an important issue. Therefore, I believe it is worth publication in Nature Communications.

There are some specific comments.

1) In figure 1(d), the replica of the electron band is seen as a trace of intensity in the original hole band at ~ 100 meV binding energy. I am a little curious why the intensity suddenly of the replica electron band drops as soon as the band goes away from the hole band. Observation of the replica electron band is the most important experimental evidence. So, it would be very nice if this part can be a little further solidified.

2) In third paragraph of the Results section, electron densities are estimated for (001) and (110) surfaces. Are they estimated from the Fermi surface area? They are more than twice different, but authors state

that k_F are similar. How can they be? Does k_F of (110) refers to the major axis of the elliptical Fermi surface in Fig. S1(b)? (*The figure is labeled as Fig. S2 by mistake)

3) Starting from line 177, the authors attribute the effective mass change to the change in e-ph coupling. While this is a reasonable assumption appears to be supported by the data in Fig. S2, the effective mass also changes quite a lot upon alkali metal dosing on bulk FeSe, which cannot be due to change in e-ph coupling (it goes to opposite direction). Therefore, other things can affect the effective mass change. I suggest authors discuss other possibilities as well.

4) The manuscript could use improvements in writing. I find mistakes at various places.

Reviewer #3 (Remarks to the Author):

The replica bands observed by ARPES previously in single-unit-cell (1UC) FeSe on STO 001 surface only are believed to conceal a potential of dramatic increase of superconductivity in this system. The paper contains very simple but important message that these replica bands are also present in 1UC FeSe on STO 110 surface, another known case of highly enhanced superconductivity. This conclusion is novel and interesting for the superconducting community, and, as far as I can judge, is well supported by presented ARPES data. The paper is also well written and I gladly recommend it for publication in Nature Communications.

REVIEWERS' COMMENTS:

Reviewer #2 (Remarks to the Author):

As I have stated in the initial review, the quality of the data and discussion in the manuscript was good. I have raised a few questions/comments and the authors provided answers to them. My questions/comments are all answered with additional figure and discussion. I have no further comments for the manuscript, and would be happy to recommend it for publication.

Point-to-Point Response to Referee Reports

Referee #1:

The manuscript is indeed very interesting. The authors convincingly demonstrated the existence of the replica bands in FeSe/STO(001) and FeSe/STO(110), which play an important role in our understanding of the mechanism of T_c enhancement in these systems as compared to simple electron doping of the single layer.

We thank referee #1 for the concise summary of our paper and the positive comments about our work.

1) One important point I would like the authors to address is the formation of the oxygen vacancies at the interface of STO and their relevance on the superconducting properties of the FeSe/STO interface. Formation of the vacancies is unavoidable in this system and the authors should comment on this.

The referee raised an excellent point. It is indeed very important to show the formation of oxygen vacancies. Before growing the film, we anneal the STO(110) substrate at 830C where we see the surface reconstruction. The evidence for oxygen vacancies created at STO(110) can be directly seen by the RHEED data, which clearly shows the (4×1) reconstruction spots. The role of oxygen vacancies to the charge transfer is still under debate. One scenario suggested that the formation of oxygen vacancy could be generalized to create the two dimensional electron gases on the surface, which in turn transfer charges to the monolayer FeSe. However, this may not be enough to cover all the charges transfer. Another scenario is about the work function difference at interface that could generate the charge transfer, which is also related to the oxygen vacancies. This topic is actively debated and needs further investigation. However, many studies have indicated that the charge transfer is an important factor in increasing the T_c , with which we agree. And we also agree that the formation of oxygen vacancies is quite important, so we have added the RHEED data before and after FeSe growth in Supplementary Fig.7, and added comments regarding this point in the discussion section.

2) Another important question concerns the position of the hole bands near the Gamma point.

I think it is important to know their positions below E_F in ARPES in both FeSe/STO(001) and FeSe/STO(110) structures. If the strong coupling mechanism of superconductivity is involved the insipient bands below E_f may play also an important role.

We thank the referee raising this important point. The oxygen exposed on different STO substrates (001 or 110) should be very different. The oxygen on the (110) surface is intrinsically less than (001). If we believe that the charge transfer are related to the oxygen vacancies. Then it is not surprising if the charge transfer at (110) is less than (001). We also observe that the position of the hole bands at (110) is closer to the E_F than (001). This result is consistent with the measurement presented by Zhang et al. (Ref. 11). The interface quality on FeSe/STO(110) is worse than (001), and also due to the matrix element effects, we are not able to resolve the top position of the hole band. However, by fitting the MDC of the band, we can estimate the position to be ~ 20 meV below E_F . We agree that the strong coupling mechanism of superconductivity may involve the insipient bands below E_F , as shown in Supplementary Figure 1. And the hole bands at Gamma point may play an important role in the electron pairing. We have added a couple sentences in the main text addressing the shift of the bands at Gamma and cite paper [Chen, X. *et al. Phys.Rev.B* **92**, 224514 (2015)] for reader to explore further.

Once these questions are addressed the manuscript could be recommended for publication

Reviewer #2 (Remarks to the Author):

Zhang et al. performed ARPES experiments on FeSe/STO(110), STO(110) and STO(001). They devised a careful experiment for a comparative study of FeSe thin film on STO(001) and STO(110). Citing the omnipresence of an energy loss mode whose energy is similar to phonon energy of ~ 100 mV, the authors argue that the phonons should play a critical role in enhancing the T_c . They provide a possible story for the strong coupling between the electron in FeSe and phonon in STO via formation of surface dipole. While observation of the mode which is likely to be from phonon does not necessarily mean the e-ph coupling is the culprit, the data and arguments are reasonable as an possible route to the enhanced superconductivity. The work further solidifies their original work that was published in Nature and clarifies an important issue. Therefore, I believe it is worth publication in Nature Communications.

We thank referee #2 for the careful review of our paper and the positive comments on our results.

There are some specific comments.

1) In figure 1(d), the replica of the electron band is seen as a trace of intensity in the original hole band at ~100 meV binding energy. I am a little curious why the intensity suddenly of the replica electron band drops as soon as the band goes away from the hole band. Observation of the replica electron band is the most important experimental evidence. So, it would be very nice if this part can be a little further solidified.

We thank the referee for raising this important point. The lower interface quality of FeSe films on STO(110) generates disorder in our films and blurs our spectra when compared to the STO(001) surface. In addition, as seen in the main electron band, intensity is suppressed near the M point, most likely due to matrix element effects. Therefore, this intensity suppression at the bottom of the band should also be present in the replica band, whose momentum location makes an illusion that the replica electron bands drops as soon as the band goes away from the hole band. The spectral weight distribution along both bands are consistent and confirms the existence of the replica band. Especially the most bright part in the main electron band (two wing-like parts) is totally consistent with the replica band even it is superposed on the relative homogeneously main hole band. The solidity of replica band can also be clearly demonstrated by the replica of the hole band.

*2) In third paragraph of the Results section, electron densities are estimated for (001) and (110) surfaces. Are they estimated from the Fermi surface area? They are more than twice different, but authors state that k_F are similar. How can they be? Does k_F of (110) refers to the major axis of the elliptical Fermi surface in Fig. S1(b)? (*The figure is labeled as Fig. S2 by mistake)*

We thank the referee for bringing up this issue. Indeed the Fermi topology of (001) and (110) is very different. We have added Supplementary Figure 3 to highlight the difference. In our measurement, we use the s -polarized light which highlights the d_{xy} bands while the $d_{xz/yz}$ bands appear much weaker. As shown in the figure above, at the (001) surface, the d_{xy} Fermi pocket is circular, while on the (110) surface, all the three orbitals show similar elliptical pockets. The band present in Fig. 2(a) and (c) is mainly from the d_{xy} band and with the similar k_F , we can

understand the difference in Fermi surface area and carrier density. The k_F of (110) indeed refers to the major axis of the elliptical Fermi surface. We have corrected the figure label.

3) Starting from line 177, the authors attribute the effective mass change to the change in e-ph coupling. While this is a reasonable assumption appears to be supported by the data in Fig. S2, the effective mass also changes quite a lot upon alkali metal dosing on bulk FeSe, which cannot be due to change in e-ph coupling (it goes to opposite direction). Therefore, other things can affect the effective mass change. I suggest authors discuss other possibilities as well.

We thank the referee for this very good suggestion. This is an interesting point to discuss in the paper as the change of effective mass is an effective way to show the strength of correlation effects. We agree that there are other possibilities that can affect the effective mass. As indicated by the referee, alkali metal doped FeSe single crystals have large changes in effective mass. This could be due to enhanced correlation effects arising from the development of a two dimensional system on top of the single crystal. And the T_c is also increases with the change of effective mass. Compare to our work, it is possible that they go to opposite direction, since the doping of electron could both increase or decrease T_c , which only depends on what kind of role the doping plays. The most direct evidence is also in the K doped FeSe experiment. A superconducting dome has been mapped at our preprint K doped FeSe manuscript (Ref. 25). We have added a discussion of the change in effective mass in the main text.

4) The manuscript could use improvements in writing. I find mistakes at various places.

We appreciate that the referee raised this point. We have made several changes to the text.

Reviewer #3 (Remarks to the Author):

The replica bands observed by ARPES previously in single-unit-cell (IUC) FeSe on STO 001 surface only are believed to conceal a potential of dramatic increase of superconductivity in this system. The paper contains very simple but important message that these replica bands are also present in IUC FeSe on STO 110 surface, another known case of highly enhanced superconductivity. This conclusion is novel and interesting for the superconducting community, and, as far as I can judge, is well supported by presented ARPES data. The paper is also well written and I gladly recommend it for publication in Nature Communications.

We thank the referee for such positive comments.

Summary of changes:

1. As suggested by the format check list, we have shortened the abstract to no more than 150 words, and remove the references mark on the abstract.

2. As suggested by referee #1, we have added RHEED data to show the formation of oxygen vacancies in the Supplementary Fig.7 and in the discussion section: “The mechanism of electron doping at the interface of FeSe/STO(001) is still actively debated^{9,24}. However the formation of oxygen vacancies may play an important role and have also been observed in FeSe/STO(110)(see Supplementary Fig. 7).” We added reference 24.
3. As suggested by referee #1, we have added a figure in the Supplementary Fig.1, and in first section of the Result we add: “Our results show the hole band at Gamma point shifts toward lower binding energy (with the band top located at ~20 meV below Fermi level) compared to FeSe/STO(001)(See Supplementary Fig. 1). It has been reported the presence of incipient bands below E_F may play an important role in the electron pairing¹⁵.” We added reference 15.
4. As suggested by referee #2, we have added a schematic figure of the Fermi surface of STO(001) and STO(110) in Supplementary Fig. 3. We have also made a note in the main text.
5. As suggested by referee #2, we added the discussion on the change of effective mass: “However, it should be noted the change of effective mass has been used as a direct tool to reflect correlation effects²². The gradual doping of K on the top of FeSe single crystal shows a heavier effective mass with a higher T_c , which has been interpreted as the enhanced correlations due to the development of a two dimensional system on top of the single crystal.”
6. We improved the writing in the manuscript. The changes have been highlighted in red.